# Induction of Adipose Tissue Browning as a Strategy to Combat Obesity

**DOI:** 10.3390/ijms21176241

**Published:** 2020-08-28

**Authors:** Alina Kuryłowicz, Monika Puzianowska-Kuźnicka

**Affiliations:** 1Department of Human Epigenetics, Mossakowski Medical Research Centre PAS, 02-106 Warsaw, Poland; mpuzianowska@imdik.pan.pl; 2Department of Geriatrics and Gerontology, Medical Centre of Postgraduate Education, 01-826 Warsaw, Poland

**Keywords:** brown adipocyte, beige adipocyte, white adipose tissue browning, thermogenesis, obesity

## Abstract

The ongoing obesity pandemic generates a constant need to develop new therapeutic strategies to restore the energy balance. Therefore, the concept of activating brown adipose tissue (BAT) in order to increase energy expenditure has been revived. In mammals, two developmentally distinct types of brown adipocytes exist; the classical or constitutive BAT that arises during embryogenesis, and the beige adipose tissue that is recruited postnatally within white adipose tissue (WAT) in the process called browning. Research of recent years has significantly increased our understanding of the mechanisms involved in BAT activation and WAT browning. They also allowed for the identification of critical molecules and critical steps of both processes and, therefore, many new therapeutic targets. Several non-pharmacological approaches, as well as chemical compounds aiming at the induction of WAT browning and BAT activation, have been tested in vitro as well as in animal models of genetically determined and/or diet-induced obesity. The therapeutic potential of some of these strategies has also been tested in humans. In this review, we summarize present concepts regarding potential therapeutic targets in the process of BAT activation and WAT browning and available strategies aiming at them.

## 1. Introduction

In a world struggling with a pandemic of obesity, there is a constant need to seek new therapeutic strategies for its treatment. The pathogenesis of simple obesity assumes an imbalance between the energy intake and expenditure that leads to the accumulation of energy surplus in the form of adipose tissue. Therefore, the available methods of obesity treatment are based on either reducing the calorie intake (diets, pharmacological approaches, bariatric surgery) or increasing energy expenditure (physical activity).

In newborns, nonshivering thermogenesis in brown adipose tissue (BAT) plays a significant role in maintaining body temperature, while in older individuals, this mechanism seemed to be less pronounced since BAT was thought to fade with age. However, the development of modern imaging techniques such as positron emission tomography (PET) has enabled the location of BAT deposits in various areas of the body of adults. It has also been found that brown adipocyte-like cells can be dispersed in white adipose tissue (WAT). Moreover, the amount of BAT positively correlates with the energy expenditure, and in obese subjects is significantly lower than in slim individuals. Therefore, strategies aimed at induction and/or activation of BAT could be potentially useful in the treatment of obesity.

In this review, we summarize present concepts regarding potential therapeutic targets in the process of BAT activation and WAT browning and available strategies aiming at them.

## 2. Types of Adipose Tissue and Their Function

Adipose tissue (AT) is an organ comprised of different cells located in the intercellular matrix. These cells include adipocytes, preadipocytes, adipose stem cells (ASCs), fibroblasts as well as endothelial, nerve and immune cells whose interactions decide about the AT homeostasis.

Traditionally, two types of adipose tissue, white (WAT) and brown (BAT), are described that differ in their origin, function, and morphology. WAT is a dominant type of AT in adult humans that can be found in different depots throughout the body. Even though WAT’s primary function is energy storage, its depots differ in their metabolic and secretory activity.

In mammals, two developmentally distinct types of brown adipocytes exist the classical, or constitutive BAT (cBAT) that arises during embryogenesis; and the beige or brite (brown-in-white) adipose tissue, that is recruited postnatally within WAT or skeletal muscle, and can be referred to as recruitable BAT (rBAT) [1,2,3,4].

cBAT is particularly well developed in infants and small children, that due to the unfavorable surface-area-to-volume ratio are at risk of hypothermia since activation of adaptive thermogenesis in cBAT is essential to maintain normal body temperature. Brown adipocytes have this ability thanks to the high content of mitochondria reach in uncoupling proteins 1 (UCP1 also called thermogenin) responsible for the uncoupling of electron transport from the production of chemical energy in the form of adenosine triphosphate (ATP). The resulting change in the balance of electrons and protons across the mitochondrial membrane leads to the loss of energy as heat. Compared to white adipocytes, constitutive brown cells are smaller and contain multiple (instead of one) lipid droplets of varied sizes. However, brown adipocytes constitute a very heterogeneous population and may vary in expression of classic marker genes such as *UCP1* as well as many others, including fatty acid glutamate, amino acid, and ion transporters [5]. In general, based on the *UCP1* expression and mitochondrial content, the population of brown adipocytes can be divided into two distinct categories with a high and low thermogenic potential. The second type of brown adipocytes is characterized by a high expression of genes related to fatty acid uptake, cell-to-cell trafficking, and UCP1-independent thermogenesis. Moreover, exposure to cold induces interconversion of low-thermogenic brown adipocytes to high-thermogenic, which can be subsequently reversed when thermoneutral conditions are restored [6].

Beige adipocytes are histologically very similar to cBAT and equivalent in their thermogenic potential (both in animals and humans); however, they express characteristic gene markers that distinguish them from cBAT and WAT adipocytes [7]. These differences refer to the developmental origins as well as response to hormonal stimuli or genetic manipulations [8].

In infants, cBAT constitutes 5% of AT and regresses with aging by transforming into WAT. However, PET scans enabled to localize several depots of cBAT also in adults. The main cBAT depot localizes in the cervical-supraclavicular region. However, lower amounts of cBAT can also be found in the peri-/suprarenal and paravertebral regions, around the major vessels, in the neck, and the mediastinum and their primary function is to generate heat to maintain core temperature [9].

The body mass index (BMI) is inversely related to BAT activity; therefore, it was plausible that impaired BAT activity contributes to obesity [10]. In clinical studies, BAT fluorodeoxyglucose (FDG) uptake was approximately four-fold higher in the lean than in the overweight/obese subjects [11]. Moreover, it is estimated that 50 g of active BAT can increase daily energy expenditure in humans by 20% [12]. With this in mind, one can suppose that an increased BAT volume could help to maintain body weight or even contribute to weight loss especially in those individuals, who after several attempts to lose bodyweight experience a fall in energy expenditure more significant than what can be accounted for by the loss of fat-free mass and fat mass [13]. These efforts could be implemented in two ways: by enhancing the activity of cBAT or by increasing the number of beige adipocytes (rBAT) [14].

## 3. Adipose Tissue Browning

The developmental origin of rBAT is distinct from cBAT. It occurs through two different pathways, depending on the fat-depot: either induction of differentiation of adipocyte progenitor cells or through transdifferentiation of mature white to brown adipocytes (and vice versa) [15].

### 3.1. Human Adipocyte Progenitor Cells

Human adipocyte progenitor cells (APCs, also known as adipose stem cells, ASCs) residents in the stromal vascular fraction of WAT can differentiate to white, brown and beige adipocytes. White adipocytes arise from APCs that do not express myogenic factor 5 (Myf-5), while brown adipocytes, from precursors expressing Myf-5-88 that have specific gene expression signatures predictive of mature adipocyte thermogenic potential [16]. The origin of beige adipocytes present in WAT is still a subject of investigation. They derive either from APCs (both existing in WAT and migrating from other tissues) or from the transdifferentiation of mature white adipocytes [17,18]. The concept that beige adipocytes can differentiate from APCs migrating from other organs it was initiated by the fact that they are in a close relationship with smooth muscle-like cells (SMCs) what is reflected by a similar gene expression signature [8].

Beige adipocytes occur in depots of white fat in response to different stimuli (e.g., cold exposure or chronic catecholamine stimulation—described in detail below) and are not derived from cell lineages expressing Myf-5 or PRDM16 (PR domain containing 16, a transcription factor that controls the development of brown adipocytes [15]. Interestingly, in vivo models of beige adipogenesis revealed a depot-selective heterogeneity of APCs in their potential to transform toward beige adipocytes, suggesting an influence of the local microenvironment [19]. Single-cell genomics studies have significantly improved our understanding of the developmental origin and regulatory pathways of beige adipocyte progenitors [20]. These studies identified CD81 (Cluster of Differentiation 81) as a highly selective marker for the beige APCs. A number of *CD81+*APCs negatively correlates with a body mass, glucose intolerance, insulin resistance, and AT inflammation in experimental models of diet-induced obesity [21].

Interestingly, induction of beiging is associated with significant alternations in WAT immune environment including the expansion of both M1 (classically activated) and M2 (alternatively activated) macrophages and a unique subpopulation of macrophages with a gene expression profile enriched for proliferation, migration, adhesion, lipid uptake and metabolism, extracellular matrix remodeling and proteolysis, and foam cell differentiation. These actions are required for beige adipogenesis, which begins with the clearance of dead white adipocytes followed by the recruitment of new APCs [22]. Moreover, a special population of macrophages involved in norepinephrine-mediated regulation of thermogenesis has been identified. These sympathetic neurons–associated macrophages (SAMs) mediate in the clearance of norepinephrine via expression of solute carrier family 6 member 2 (SLC6A2, norepinephrine transporter) and monoamine oxidase A (MAOA, a degradation enzyme). Local ablation of *Slc6a2* in SAMs results in WAT browning, expansion of BAT, and an increase in thermogenesis, leading to significant weight loss in animal models of obesity. Since human sympathetic ganglia also contain SAMs with analogous molecular machinery for norepinephrine, they may constitute a potential target for obesity treatment [23].

Several lines of evidence suggest that WAT browning also depends on proper eosinophil-derived cytokines activity (namely interleukin (IL) 4/IL-13 signaling), which is necessary for the alternative activation of anti-inflammatory M2 macrophages [24]. The thermogenic challenge is also associated with the expansion of adaptive immune cell populations and upregulation of interleukin 10 production, which antagonizes adrenergic-mediated thermogenesis, providing a mechanism whereby the immune system helps control the consumption of energy reserves [25].

Apart from the differentiation from APCs, beige adipocytes can arise due to transdifferentiation from white fat cells. This theory was supported by findings that during cold stimulation, an intermediate adipocyte phenotype between the typical brown and white morphology can be observed, and the number of cells remains constant [18,26]. Moreover, the ability of white adipocytes to transdifferentiate into beige cells can be reversible. For instance, in C57BL/6 mice, a one-week cold stimulation at 8 °C induces expression of the brown/beige-specific genes in white adipocytes reflected by the changes in tissue morphology (increase in the number of cells with a multilocular brown phenotype), that is completely reversible within 5 weeks of warm adaptation. Interestingly, this kind of conversion is not observed in classical brown adipocytes after the cold stimulation is over [14]. It was found that exposure to temperature challenge causes chromatin reprogramming in beige (but not brown) adipocytes leading to retaining an epigenomic memory. Histone methylation is a proven mechanism mediating in cellular memory to induce and maintain the beige adipocyte characteristics [27]. Due to this phenomenon, beige cells, when they have previously been exposed to cold once, can quickly reboot their thermogenic machinery when re-challenged [28].

### 3.2. Mechanism Involved in Adipose Tissue Browning and Activity

The transcriptional control of brown adipocyte formation has been analyzed extensively and reviewed in great detail elsewhere [29]. The proliferator-activated receptor gamma (PPARγ) and the CCAAT/enhancer-binding proteins (specifically C/EBPα/β/δ) transcription factors are essential elements of the transcriptional cascade that precedes the formation of mature, fully differentiated adipocytes. These transcription factors are involved in the differentiation of both brown and white adipocytes. Initialization of the brown adipogenic program requires the PPARγ coactivator 1α (PGC-1α, activated by p38 mitogen-activated protein kinase—MAPK, and activating transcription factor 2—ATF2) and PDRM16 transcription factor which interacts with C/EBPβ. PRDM16 plays a pivotal role in maintaining the beige adipocyte phenotype, and when its expression is low, beige adipocytes may turn into white adipocytes again [30].

Several modulators of PPARγ binding activity may determine differentiation towards brown adipocytes. Among them is sirtuin 1 (SIRT1, the NAD-dependent deacetylase), which deacetylates PPARγ and thereby enables recruitment of PRDM16 and initiation of the brown fat specific program [31]. Another example is a transcription factor EBF2 (early B-cell factor 2) responsible for the PPARγ recruitment to genes crucial for the brown adipogenic processes [32]. PPARγ is crucial for *UCP1* transcription during brown adipocytes differentiation; however, it is repressed in mature brown adipocyte, where a dominant role in the regulation of *UCP1* transcriptional activity plays another member of the PPAR family—PPARα which also regulates the activity of genes involved in lipid oxidation [33].

Adrenergic stimulation is crucial for the initiation of thermogenic pathways, and sympathetic nerves abundantly innervate both BAT and WAT. Different stimuli, like, e.g., abovementioned exposure to cold, lead to increased noradrenaline release and subsequent stimulation of various subtypes of β-adrenergic receptors (ADRBs), leading to the proliferation of brown adipocytes and activation of lipolysis and/or of thermogenesis. ADRB3 is the primary receptor involved in pathways related to adaptive thermogenesis in brown or beige adipocytes [34].

AMP-activated protein kinase (AMPK) also plays an essential role in regulating the sympathetic stimulation of WAT browning and BAT thermogenesis. In response to a negative energy balance, AMPK switches the intracellular processes to those aiming at ATP production. Hypothalamic inactivation of AMPK results in enhancement of WAT browning and BAT thermogenesis, and therefore AMPK represents a potential target for obesity treatment (see following sections) [35]. In turn, foxhead P1 (FOXP1) acts as a negative regulator of brown/beige adipogenic differentiation directly repressing the transcription of *ADRB3*. Adipose-specific deletion of *FOXP1* leads to an increase in BAT activity and WAT browning. Inversely, overexpression of *FOXP1* in adipocytes impairs adaptive thermogenesis and promotes diet-induced obesity [36].

Adrenergic stimulation is responsible for rapid responses of BAT to external stimuli, but triiodothyronine (T3) by increasing the capacity of cells to respond to catecholamines also plays an essential role in the regulation of thermogenesis. Cold-induced noradrenergic stimulation activates in BAT the type 2 5′-iodothyronine deiodinase (DIO2), catalyzing the conversion of thyroxin (T4) to T3. T3 acting by its α and β nuclear receptors (TRα and TRβ, respectively) increases, among others, the expression of genes encoding UCPs. What is important human obesity is associated with decreased expression of above-mentioned thermogenesis-related genes in WAT that can reduce its reactivity to both hormonal and adrenergic stimuli involved in activation of thermogenesis [37].

While adrenergic stimulation plays a central role in the induction of adipose tissue browning, and activation of thermogenesis, activation of gamma-aminobutyric acid (GABA) signaling exerts opposite effects in adipocytes. The influence of GABA on energy homeostasis is composed. On the one hand, the activation of the GABA-A receptor in the hypothalamus suppresses food intake and reduces body weight [38]. On the other hand, the Agouti-related peptide (AgRP) and neuropeptide Y (NPY) neurons—both stimulating appetite, are mostly GABAergic [39]. In the context of brown adipocytes, the constitutive activation of the GABA-BR1 receptor in obese mice leads to excessive calcium influx into the mitochondria causing mitochondrial calcium overload and oxidative stress that is accompanied by low UCP1 expression. Moreover, in experimental animals, high fat diet leads to a chronic up-regulation of the GABA-BR1 in adipose tissue and its translocation into cellular membrane, making adipocytes of obese animals more sensitive to the damaging effects of GABA. In turn, suppression of excessive signaling by the inhibition of mitochondrial calcium overload restores normal BAT function [40].

Over the past decade, many other pathways involved in the development of brown adipose tissue and the regulation of thermogenesis have been discovered. The examples are atrial (ANP) and brain (BNP) natriuretic peptides. Both of them, via activation of natriuretic peptide receptors (NPRs) and downstream mediators such as cyclic guanosine monophosphate (cGMP), protein kinase G (PKG) and mammalian target of rapamycin complex 1 (mTORC1), play a central role in the regulation of fluid balance and hemodynamic. However, identification of NP receptors (NPRs) expression in adipose tissue shed new light on ANP/BNP pathways’ role in the whole-body homeostasis. Subsequently, ANP was found to induce lipolysis in white adipocytes and thermogenesis in BAT, while both NPs, via the cGMP-PKG-mTORC1 pathway, can increase UCP1 expression and mitochondrial content in adipocytes leasing to WAT browning [41,42,43]. Accordingly, mice with *npra* knockout suffer from cardiac hypertrophy and have significantly higher adipose tissue content than wild-type animals [41]. On the contrary, patients with chronic heart failure that are exposed to persistently elevated NPs levels are often diagnosed with lipodystrophy and cachexia (known as cardiac cachexia) [44].

Another example is orexins (OXs) A and B—neuropeptides that regulate sleep-wake cycles, physical activity, and appetite, but also may increase adipose tissue browning and thermogenesis [45]. Orexins, via activation of MAPK and BMPR1A signaling, are able to initiate the browning program in mesenchymal progenitor stem cells, embryonic fibroblasts, and brown preadipocytes. Subsequently, mice deficient with orexins have impaired differentiation of brown preadipocytes toward mature brown cells and decreased adaptive thermogenesis. Therefore, in response to the high caloric diet, they experience a rapid increase in adipose tissue volume [45]. Interestingly, a decrease in BAT’s orexin receptors expression, leading to the loss of its thermogenic capacity, was found to be responsible for the aging-associated increase in adiposity in mice [46]. Moreover, low orexin serum levels are associated with obesity in humans [47].

### 3.3. Role of Metaflammation in the Regulation of WAT Browning and BAT Activity

Both in humans and experimental animals, obesity is associated with permanently elevated serum levels of inflammatory markers. This chronic, low-grade inflammatory state associated with excessive adiposity is called metaflammation. The pathogenesis of metaflammation is composed and still an object of investigations. Briefly, overnutrition leads to the excess of free fatty acids (FFA) in the circulation that, together with the accumulation of lipids in tissues, cause profound changes in cell metabolism. These include, among others, mitochondrial dysfunction, endoplasmic reticulum stress, and hypoxia resulting in increased expression of genes encoding cytokines, chemokines, and adhesion molecules that subsequently attracts infiltrating immune cells that additionally contribute to the synthesis of pro-inflammatory mediators impairing tissue function [48]. Even though WAT is believed to be a primary site of metaflammation, the inflammatory process related to obesity (assessed by the intensity of macrophages infiltration and concentration of pro-inflammatory cytokines) also affects BAT, as it was shown in animal models [49,50]. Inflammatory mediators impair BAT function that concern also its thermogenic activity. In genetically obese *ob*/*ob* mice and mice with diet-induced obesity, high concentrations of inflammatory markers (like e.g., tumor necrosis factor α or IL-1β) inhibit activation of PPARγ, leading to the decreased expression of UCP1, that translates to impaired cold-induced thermogenesis. Interestingly, this process can be reversed by the depletion of pro-inflammatory macrophages [51]. Moreover, it was shown that activation of toll-like receptors (TLRs, that by recognition of conserved pathogen-associated molecular patterns represent the first line of defense against infections), or direct administration of pro-inflammatory cytokines, can repress induction of WAT browning induced by activation of ADRBs [52].

The nuclear factor κB (NF-κB) cascade is a chief intracellular signaling pathway involved in the development of inflammatory response in the adipose tissue. The NF-κB family of transcription factors consists of five members: NF-κB1, NF-κB2, RelA, RelB, and c-Rel that act as homo- and heterodimers. NF-κB dimmers are sequestered in the cytoplasm due to their association with κB inhibitors (IκBs). Various stimuli, including those transmitted by TLRs, cause phosphorylation of IκBs and subsequent dissociation from NF-κB. Dissociation of IκB exposes the nuclear localization signal in NF-κB transcription factors and leads to their translocation to the nucleus. There, NF-κB binds to the promoters of target genes, including those involved in the inflammatory response: cytokines, chemokines, adhesion molecules, and many others that inhibit browning of WAT [53].

Both FFA and the pro-inflammatory cytokines can activate c-Jun N-terminal kinase (JNK), leading to increased phosphorylation of interferon regulatory factor 3 (IRF3) and in this way contributing to the reduced UCP1 expression and WAT browning [54]. Moreover, activated JNK exacerbates metaflammation by increasing expression of pro-inflammatory cytokines in adipocytes and polarization of macrophages toward the M1 pro-inflammatory phenotype [55].

Inflammation also impairs sympathetic signaling in BAT. The suggested molecular mechanisms underlying this phenomenon include, among others, reduced cAMP synthesis due to the chronic activation of NF-κB cascade and increased clearance of norepinephrine by the obesity-specific population of neuron-associated macrophages. Both result in repressed BAT thermogenic activity and WAT browning [23].

In summary, even though WAT is the primary side of metaflammation, the chronic inflammatory state has deleterious effects on the thermogenic function of BAT. Moreover, local pro-inflammatory signaling in white adipose tissue impairs its browning.

Understanding of the mechanisms involved in WAT browning and BAT activation allowed for the identification of critical molecules and critical steps of both processes and, therefore, many new therapeutic targets. Several non-pharmacological approaches, as well as chemical compounds aiming at the induction of WAT browning and BAT activation, have been tested for their utility for obesity treatment in vitro as well as in animal models of genetically determined and/or diet-induced obesity. The therapeutic potential of some of these strategies has also been tested in humans. Figure 1 summarizes selected molecular mechanisms involved in WAT browning and BAT activation that could constitute potential therapeutic targets for obesity treatment.

## 4. Non-Pharmacological Interventions Aiming at White Adipose Tissue Browning and Brown Adipose Tissue Activation

### 4.1. Cold Exposure

Chronic exposure to cold results in the activation of sympathetic neurons and the release of norepinephrine through the activation of ADRB (especially ADRB3) and downstream cyclic adenosine monophosphate (cAMP) cascade turns on a thermogenic gene program that defines a BAT-like molecular phenotype [56]. Accordingly, denervation of WAT leads to an increase in adipocyte number and volume in experimental animals [57]. The density of WAT sympathetic innervation depends on its anatomical location, and cold exposure causes an increase in the number of sympathetic noradrenergic nerve fibers, which is associated with the emergence of beige adipocytes [58].

In humans, daily cold exposure can increase BAT volume even by 45% and oxidative metabolism by 182% [59]. This phenomenon concerns also browning of human WAT (the direct exposure of primary white adipocyte cell cultures to 16 °C for 30 min leads to an increase in thermogenic gene expression), and has seasonal variations since the expression of thermogenic markers in WAT is significantly increased in biopsies obtained during winter compared to summer [60].

There is also evidence from patients with pheochromocytoma that the prolonged elevation of norepinephrine levels plays a central role in the induction of WAT browning; however, the scale of this phenomenon has to be determined. In a study by Frontini et al., 50% of pheochromocytoma patients had UCP1-positive brown adipocytes in the omental WAT, while Vergnes et al. found brown adipocytes only in periadrenal but neither omental nor subcutaneous AT [61,62].

### 4.2. Diet

In vitro and in vivo data suggest that dietary components can increase thermogenic capacity by augmentation of BAT activity, stimulation of white adipocyte transdifferentiation into beige cells, and differentiation of APCs into beige adipocytes. The molecular mechanisms triggered by dietary molecules to induce thermogenic responses include, but are not limited to, activation of ADRB3 and AMPK/PGC-1α signaling, stimulation of thermogenic and anti-inflammatory cytokine secretion as well as epigenetic changes.

Capsaicin and its analogs capsinoids (found, e.g., in red pepper) are probably the most extensively studied food components involved in BAT activation. Their main mechanisms of action include direct activation of ADRB3 and indirect—via enhancing catecholamine secretion from adrenals and subsequent activation of the sympathetic nervous system [63]. Administration of capsinoids (30–300 mg/kg body weight) to rats results in time- and dose-dependent increase in inguinal BAT sympathetic nerve activity and browning of WAT. The transient receptor potential vanilloid (TRPV) 1 located on primary afferent neurons seems to be a primary target for capsinoids. Its activation is associated with catecholamine production and SIRT1-mediated deacetylation of PRDM16 [64]. The same effect was observed in C57BL/6 mice, where combined treatment of mild cold exposure and capsinoids via increased browning of WAT was able to counteract diet-induced obesity [65]. Both capsaicin and capsinoids can enhance thermogenesis in humans, too [66]. A combination of capsinoids (9 mg/d) with cold exposure for 6 weeks stimulated BAT activity and reduced adiposity, even in individuals with low initial BAT function [67]. Several other dietary compounds (such as menthol, 6-paradol, allyl isothiocyanates, and benzyl isothiocyanates) can increase thermogenesis via activation of TRPV [68]. However, before these compounds are used in the treatment of obesity, it should be verified if the long-term stimulation of the sympathetic nervous system associated with their use does not cause adverse side effects.

Another representative of dietary compounds with thermogenic potential is resveratrol (3,5,4′-trihydroxy-trans-stilbene, RSV), a polyphenol present in grapes, berries, peanuts, and selected teas. The underlying mechanism includes activation of SIRT1, that by direct deacetylation of PPARγ recruits BAT program coactivator PRDM16 resulting in induction of genes typical for BAT and downregulates genes that are specific for WAT in 3T3-L1 preadipocytes [69]. Moreover, the addition of RSV to a cell culture of differentiated inguinal WAT-derived stromal vascular cells significantly increased brown and beige adipocyte markers’ expression. This effect of resveratrol on beige adipocyte formation is partially mediated by phosphorylation of AMPK since AMPK inhibition eliminates the browning effects of RSV on WAT [70]. Besides, a recent study pointed on the mammalian target of the rapamycin (mTOR) pathway to be involved in RSV induced WAT browning [71].

In animal models of obesity, the addition of RSV to a high-fat diet-induced UCP1 expression and oxygen consumption in BAT, resulting in the increased basic metabolic rate and, in this way, successfully counteracted accumulation of AT [72]. Moreover, the upregulation of SIRT1 by RSV results also in the repression of NF-κB pro-inflammatory responses in adipocytes and macrophages that infiltrate adipose tissue, leading to an improvement in insulin signaling and insulin resistance, as it was shown in genetically obese db/db mice [73]. Interestingly, a four-week treatment with 500 mg of trans-resveratrol led to the upregulation of thermogenesis-related genes, also in human WAT [74]. This finding can be partially responsible for the beneficial influence of RSV and its derivates on weight management observed in clinical trials [75].

What is important, after ingestion, RSV is metabolized in the colon by the gut microbiota, and part of the beneficial effects of RSV is mediated via its microbial metabolites. Therefore, depletion of the gut microbiota may counteract the beneficial effects of RSV on brown adipose tissue and glucose metabolism [73].

Curcumin is a naturally occurring curcuminoid of turmeric, with anti-obesity and anti-hyperglycemic properties. Administration of curcumin, by arresting lipogenesis in the liver and the inflammatory pathways in adipocytes, prevents the development of high-fat diet-induced obesity and insulin resistance in rodents [76]. In turn, in a randomized, controlled study, in overweight subjects with central adiposity and glucose intolerance, adding curcumin to nutritional intervention and physical activity significantly increased weight and body fat reduction [77].

Moreover, due to the ability to activate PPARγ and AMPK, curcumin also has thermogenic properties. The browning potential of curcumin has been shown in vitro in 3T3-L1 cells and mouse primary white adipocytes [78]. In this study addition of curcumin to cell cultures induced the beige phenotype and drove the BAT thermogenic program through significant upregulation of brown fat-specific genes. Moreover, curcumin increased the number of mitochondria and expression of critical mitochondrial proteins such as carnitine palmitoyltransferase I (CPT1, responsible for the oxidation of fatty acids in brown adipocytes) and cytochrome C that plays a key role in mitochondrial oxidative phosphorylation. Therefore, the anti-obesity properties of curcumin are based on both: ability to inhibit lipogenesis and induction of thermogenesis and adipose tissue browning [78].

This observation has been confirmed in vivo when curcumin administered 50–100 mg/kg/day decreased body weight and fat mass without affecting food intake but by increasing energy expenditure and body temperature in mice. These curcumin effects were associated with the emergence of beige adipocytes with an increase of brown-specific markers expression and mitochondrial biogenesis in WAT. Moreover, treatment with curcumin reduced macrophages infiltration and proinflammation cytokine expression in inguinal WAT, suggesting its potential to counteract obesity-associated inflammation [79,80]. However, curcumin’s effectiveness in the induction of adipose tissue browning in humans has to be established yet.

Both animal and human studies suggest that consumption of green tea catechins (GTC, e.g., epigallocatechin, epicatechin gallate, and epicatechin) is associated with an increase in AT metabolism and energy expenditure. Administration of GTC to rats on a normal fat diet reduced perirenal WAT weight that was associated with the up-regulation of UCP1 expression in BAT, suggesting that catechins’ suppressive effect on body fat accumulation is associated with increased BAT activity [81]. A meta-analysis of placebo-controlled studies revealed that intake of catechin-caffeine mixtures significantly accelerated energy expenditure and fat oxidation in humans [82], while regular ingestion of a catechin-rich (540 mg/day) beverage increased BAT density and decreased extramyocellular lipids content [83]. These catechins’ actions are probably mediated by modulation of PPARγ levels in AT [84]. It was also suggested, that GTC may stimulate thermogenesis and fat oxidation through the inhibition of catechol *O*-methyltransferase, leading to activation ADRB3 signaling [68]; however, this concept was not confirmed in a clinical trial, where consumption of catechins did not influence *ADRB3* expression in human AT [85].

Berberine is a plant-derived compound, used as an anti-diarrhea drug. In clinical trials, berberine was found to have an antidiabetic and antihyperlipidemic effect, as well as favorably influence body weight and liver fat content in patients with non-alcoholic fatty liver disease (NAFLD) [86]. In vitro and animal studies suggest that these effects are mediated via peripheral activation of AMPK-PGC-1α pathway, resulting in the development of beige adipocytes in WAT and increased expression of thermogenic genes and mitochondrial content in BAT and primary adipocytes, however the exact molecular mechanism involved in this process remains unknown [87]. The thermogenic potential of berberine was also demonstrated in humans: 1-month administration of berberine to mildly overweight patients increased BAT mass (due to promotion of brown adipocyte differentiation) and activity that resulted in body weight and improvement in insulin sensitivity [88].

Fish oils, rich in the omega-3 polyunsaturated fatty acids (n-3 PUFA), represent dietary compounds of animal origin with thermogenic properties. Eicosapentaenoic acid (EPA) significantly increases the expression of thermogenesis-related genes (e.g., UCPs) and genes involved in mitochondrial biogenesis (e.g., PGC-1α) and oxidation in cultures of primary subcutaneous murine adipocytes as well as differentiated human white adipocytes and beige cells [89,90]. Subsequently, in rodents, 3–4-week supplementation with 20–30% fish oil results with significantly increased mitochondrial and thermogenic activity in BAT that translates to resistance to diet-induced obesity and improved metabolic profile [68]. However, the influence of PUFA on WAT browning can be suspended by a high-fat content in a diet [91]. The underlying molecular mechanisms responsible for the thermogenic properties of PUFA include activation of TRPV1, leading to ADRB3 stimulation but also FFAR4 (free fatty acids receptor 4 also known as G-coupled protein receptor 120—GPR120) that leads to upregulation of several miRNAs promoting adipocyte browning [92]. However, if dietary n-3 PUFAs exert thermogenic effects in humans has to be determined.

All-trans retinoic acid (ATRA) is an active metabolite of vitamin A that acts via nuclear retinoic acid receptors (RARs). RARs upon heterodimerization with retinoid X receptors (RXRs) bind with regulatory regions of target genes and, in this way, control their transcription n [93]. In vitro, ATRA induces expression of UCP1 in mice white and brown adipocytes but inhibits or has no effect on the expression of thermogenic markers in human adipocyte cell lines and primary human white adipocytes [94]. Subsequently, supplementation with ATRA (10–100 mg/kg) enhances WAT browning leading to body weight and fat mass reduction in obese rodents that is associated with a significant increase in thermogenic gene expression. Inversely, a vitamin A deficient diet increases body weight, whole-body fat mass, and reduces BAT thermogenic activity [95]. Unfortunately, ATRA thermogenic potential is significantly lower in human tissues; therefore, its application to combat human obesity seems not to be reasonable.

In summary, despite the mounting evidence from in vitro and animal studies, the thermogenic potential of several dietary compounds in humans has to be verified in clinical, placebo-controlled trials.

### 4.3. Physical Activity

Exercise and physical activity have a well-established role in the preservation of cardiometabolic health. Physical activity also elicits many benefits in AT. These include a reduction in adipocyte size and AT inflammation (e.g., by downregulation of the TLR4 ligation and subsequent activation of NF-κB pathway) as well as an increase in vascularity and mitochondrial biogenesis that results in the improvement of oxidative capacity. Moreover, physical activity is associated with an increase in beige adipocytes’ number and activity in WAT. It has been suggested that exercise-induced lipolysis increasing circulating non-esterified fatty acids (NEFA) creates a need for other areas of oxidation and that WAT browning provides these sites in order to maintain NEFA flux [96]. The exact molecular mechanisms underlying this process remain mostly unknown; however, the mediators secreted during exercise by skeletal muscle, AT, and potentially the liver in an endocrine and/or paracrine manner, occurred to play a significant role. These properties have, for instance, irisin (the cleavage product of the membrane-bound protein fibronectin type III domain-containing protein) secreted by muscles during exercise. Irisin promotes the expression of thermogenesis-related genes in primary white adipocytes cultures and stimulates BAT development and energy expenditure in rodents. These actions are mediated via the ability of irisin to recruit PGC1-α [97]. However, in contrary to animal experiments, results of human studies assessing the influence of irisin on metabolic health are not univocal, especially, that obese individuals occurred to have significantly increased irisin serum levels [98]. Moreover, several studies failed to show an increase in systemic irisin concentrations after acute or chronic endurance training in humans [96].

Interleukin-6 (IL-6) is another example of exercise-induced cytokine involved in WAT browning. Treatment of with IL-6 stimulates UCP1 expression in white adipocytes in vitro and in WAT in vivo. Subsequently, in mice with IL-6, knockout exercises do not result in the induction of thermogenic gene expression in white adipocytes [99]. Stimulation of WAT browning is also a feature of fibroblast growth factor 21 (FGF21). Moreover, FGF21 was found to facilitate the catabolism of fatty acid oxidation in WAT and energy dissipation in BAT. In mice with diet-induced obesity, the administration of FGF21 results in a reduction in body weight and whole-body fat mass due to marked increases in total energy expenditure and physical activity levels associated with the improvement of several metabolic parameters [100]. In turn, FGF21 knockout reduces the ability to adapt to chronic cold exposure with diminished browning of WAT in mice [101]. In humans, FGF21 serum levels correlate with BAT activity determined by FDG uptake [102]. However, whether the induction of WAT browning and BAT thermogenesis is a causative mechanism responsible for the beneficial effects of FGF21 on glucose homeostasis, lipid metabolism, and body weight remains controversial [103].

Additionally, β-aminoisobutyric acid (BAIBA), via activation of PPAR, occurred to act as an inductor of WAT browning both in vitro and in vivo. BAIBA is considered a myokine since it is a product of valine catabolism generated in the muscle during exercise. In epidemiological studies, plasma valine concentration correlates with obesity and hyperinsulinemia. Therefore, its utilization in working muscle can be partially responsible for the beneficial metabolic effects of exercise. Subsequently, in genetically obese mice (*ob/+*), treatment with BAIBA led to the reduction of weight gain that was accompanied by the improvement of glucose tolerance, enhanced fatty acid oxidation in BAT, and reduced lipogenesis in the liver [104].

It should also be mentioned that in animal studies, the influence of exercise on WAT browning was depot-specific (for instance concerned inguinal WAT only), which might explain discrepancies between results obtained in different studies performed humans as well as between animal and human studies. Therefore, future human studies should be designed to examine the response to exercise at multiple sites in the same adipose depot and across different depots [96]. Moreover, most available studies were performed on either sedentary or exercise-trained populations, while little is known whether regular physical has thermogenic effects on AT.

While studies on the influence of exercise on BAT formation and activity in humans are controversial, it seems that a combination of physical activity with cold exposure could be beneficial in promoting weight loss, beige fat formation, and blood lipid markers indicating cardiovascular health. This finding is of special importance since cold exposure alone, via the promotion of higher cholesterol and triglyceride levels, can have detrimental cardiovascular effects [105].

### 4.4. Microbiome

The intestinal microbiome also occurred to play a role in the regulation of WAT browning [106]. In animal models of obesity and lean animals, depletion of microbiota via activation of type 2 cytokine signaling stimulates beige adipocytes development in WAT, leading to improved glucose tolerance and insulin sensitivity [106]. Moreover, some of the abovementioned stimuli induce browning of WAT by influencing the microbiome. For instance, exposure to cold leads to the alternation of the microbial composition (e.g., increased amounts of Adlercreutzia, Mogibacteriaceae, Ruminococcaceae, and Desulfovibrio and reduced levels of Bacilli, Erysipelotrichaceae, and the genus rc4-4) in the gut in animal models and transfer of microbiota from cold-adapted mice induces browning of WAT in germ-free animals that together with activation of BAT, improves their cold tolerance, glucose tolerance and insulin sensitivity and leads to weight reduction. However, during prolonged adaptive mechanisms maximizing caloric uptake and increasing intestinal absorption, the bodyweight loss is attenuated [107,108]. In turn, exercise can promote the proliferation of specific bacterial strains, including those of Veillonella [109]. A combination of physical activity with cold exposure occurred to promote more extensive changes in the gut microbiome and subsequently increased WAT browning ant weight loss more effectively than exercise or cold exposure alone [110]. In turn, mice challenged with caloric restriction or intermittent fasting (both associated with excessive WAT browning) have also been reported to have alternations in intestinal microbiota. Subsequently, transplantation of their intestinal flora increases the number of beige adipocytes and improves insulin sensitivity in germ-free animals [111].

Even though the exact pathways stimulated in response to cold, exercise and caloric restriction have yet to be determined, increased lactate production by the microbiota (or in case of physical activity—increased lactate transport through the intestinal wall) seems to be a chief mechanism underlying this phenomenon [112]. Lactate was found to significantly induce thermogenic gene expression in vitro in murine and human lines of adipocytes [113]. Moreover, cold, exercise, and caloric restriction can significantly reduce the expression of the critical bacterial enzymes necessary for the lipid A biosynthesis, a critical component of lipopolysaccharide (LPS). Decreased LPS production led to anti-inflammatory M2 macrophage polarization and eosinophil infiltration in WAT, associated with the thermogenic reprogramming of WAT [111,114].

### 4.5. Adipose Tissue Grafting and Bariatric Surgery

BAT grafting into the visceral cavity has been successfully applied in mice. It resulted in a dose-dependent improvement of glucose tolerance, insulin sensitivity, lower body weight, decreased fat mass, and a complete reversal of high-fat diet-induced insulin resistance. BAT-derived IL-6 seems to be a chief mediator of these phenomena since the improved metabolic profile was lost when the BAT used for transplantation was obtained from IL-6 knockout mice [115].

The concept of autologous beige adipocytes transplantation has also been tested. In their study Blumenfeld et al. cultured fragments of murine WAT for 3 weeks in browning conditions, next re-implanted them to finally explant them again 8 weeks later. Tissues that were ex vivo treated with browning media showed sustained UCP1 expression, whereas the tissues cultured in control media retained a WAT-like appearance. Moreover, mice with diet-induced obesity, 4 months after injection with converted beige adipocytes, had lower body weight than those injected with control WAT. Importantly, the ex vivo conversion of white adipocytes into UCP1-expressing beige cells was also achieved in human tissues [116].

Since most of the non-invasive methods of obesity treatment have limited effectiveness from a long time perspective, bariatric surgery is nowadays the most effective method to obtain a sustained weight loss. Enhancement of WAT browning and BAT thermogenesis are mentioned as two possible mechanisms contributing to the beneficial influence of bariatric surgery on weight loss and metabolic health. Interestingly, rodent and human studies suggest that the effect of bariatric surgery on AT browning may depend on the procedure: while Roux-en-Y gastric bypass (RYGB) mainly enhances beige cell recruitment, and vertical sleeve gastrectomy (VSG) predominantly increases BAT thermogenesis. Moreover, the thermogenic effect depends on the fat content in the diet and is absent if postoperatively, a low-fat diet is applied [117].

Bariatric surgery leads to profound changes in gut enzymes and hormone secretion and intestinal microbiome composition. In case of VSG, it is believed that postsurgical enhancement in BAT thermogenesis is obtained due to the increased concentration of circulating bile acids that activate membrane-bound G protein-coupled bile acid receptor (GPBAR-1) in brown adipocytes [118]. However, this hypothesis should be supported by experiments mice with BAT-specific GPBAR-1 knockout. Present evidence does not support the role of gut hormones in the induction of post-bariatric surgery thermogenesis [117]. However, the postoperative changes in the intestinal microflora can contribute to bariatric surgery’s beneficial effects, especially that dysbiosis induced by antibiotic exposure was found to attenuate weight loss and metabolic improvement following VSG [119].

However, it should be underlined that the postsurgical increase in BAT thermogenesis and WAT browning are less markedly indicated in humans than in rodents [117]. Moreover, it is not clear to what extend the thermogenic effect of bariatric surgery contributes to the postoperative changes in weight and metabolic parameters. To clarify this issue, the effectiveness of bariatric surgery should be tested on UCP1-deficient animals; however, other, UCP1-independent mechanisms may play a dominant role in the observed phenomena [120].

The described above non-pharmacological approaches to induce WAT browning and BAT thermogenesis are summarized in Table 1.

## 5. Pharmacological Interventions Aiming at White Adipose Tissue Browning and Brown Adipose Tissue Activation

Understanding the mechanisms responsible for AT browning allowed for the development of new targeted therapies to intensify this process to treat obesity and/or improve the metabolic profile. Moreover, it allowed applying for this purpose previously known drugs used in the treatment of other diseases.

### 5.1. Adrenergic Receptors Beta Agonists

ADRBs are members of the G protein-coupled receptor family that, after activation, induce the adenyl cyclase leading to the increase in intracellular cAMP levels. cAMP acts as a second messenger activating protein kinase A that subsequently results in the phosphorylation of multiple targets, including PGC-1α and PPARs. In AT, this process may result in adipocyte proliferation and induction of nonshivering thermogenesis (BAT) and the mobilization of stored fatty acids for lipolysis (WAT). Therefore, ADRBs seem to be promising targets for obesity treatment. However, systemic activation of β-adrenergic receptors by nonselective agonists (ephedrine or isoprenaline) results in generalized energy consumption by several organs (e.g., the heart, liver, and skeletal muscle) with little effect on BAT thermogenesis or WAT browning.

Moreover, nonselective ADRBs agonists (e.g., norepinephrine), despite their thermogenic potential, can have detrimental effects on the cardiovascular system [121]. Therefore, there was a need to develop novel, selective ADRBs ligands aimed to stimulate brown adipocytes proliferation and activation. CL-316243—a selective ADRB3 agonist was found to increase *UCP1* mRNA synthesis in APCs and their differentiation towards brown adipocytes in vitro and induce WAT browning in rodents [19,122]. Recently another selective ADRB3 agonist—mirabegron, has been shown to increase BAT metabolic activity and energy expenditure in humans; however, the compound did not influence WAT browning [123]. In conclusion, despite promising preclinical data, selective ADRB3 agonists occurred to have limited efficacy in induction WAT browning in humans, probably due to the lower number of ADRB3 in human AT than in transfected cell-lines used for the in vitro experiments as well as different browning potential of WAT in rodents [124]. The lower expression of *ADRB3* in AT of obese subjects may constitute another reason for the low efficacy of these compounds in obesity treatment [37].

### 5.2. PPAR Modulators

Activation of PPARγ by synthetic ligand rosiglitazone induces expression of genes specific for brown adipocytes and mitochondrial biogenesis in white adipocyte cell lines and murine WAT through the activation of the PRDM16 pathway [2,125]. In addition, treatment with partial PPARγ agonist (GQ-16) reduced high fat diet-induced weight gain in mice that was accompanied by reduced epididymal fat mass, reduced liver triglyceride content, morphological signs of increased BAT activity, as well as, increased expression of thermogenesis-related genes in interscapular BAT and epididymal WAT [126]. Consistently, AT in mice with PPARγ knockout has reduced capacity to generate ATP that translates to a lower metabolic rate [127]. Even non-classical PPARγ agonist, imatinib that activates the receptor by blocking its phosphorylation, has also been shown to induce WAT browning in animal models [128]. However, the administration of another PPARγ agonist (pioglitazone) to obese diabetic individuals did not increase energy expenditure and weight reduction [129]. Moreover, a clinical trial treatment with pioglitazone resulted in a significant reduction of cold-induced and total BAT and increased total and lean body mass compared with placebo [130].

In preclinical studies, PPARα agonist—fenofibrate was found to induce beige cell depots in the WAT of diet-induced obese mice that were associated with higher expression of several thermogenesis-related genes with a consequent reduction in the body mass and hepatic steatosis [131]. Similar results were obtained during treatment with pemafibrate, which, besides, increased thermogenesis in BAT [132]. Co-administration of fenofibrate successfully prevented rosiglitazone-induced body weight gain in genetically obese mice; nevertheless, prolonged treatment with fenofibrate has been not reported to influence body weight and BAT content in humans [133].

### 5.3. AMPK Modulators

AMP-activated protein kinase (AMPK) plays a central role in cellular energy homeostasis acting as a central energy sensor and is expressed in AT as well as in many other tissues, including the liver, hypothalamus, and skeletal muscle. Apart from being implicated in diverse biological processes, AMPK plays a critical role in regulating fatty acid metabolism, thermogenesis, and the development of AT. Upon activation, AMPK inhibits fatty acid synthesis and stimulates their oxidation (in the muscles and the liver) and modulates adipocyte lipogenesis and lipolysis. Its activators (e.g., A-769662) enhance fatty acid oxidation leading to decrease in plasma and liver triglyceride levels and reduced body weight gain, enhanced cold-induced thermogenesis and induced browning in the inguinal fat depot in genetically obese *ob/ob* mice [35,134]. In turn, the above-mentioned natural compounds curcumin and berberine were found to activate AMPK-PGC-1α pathway increasing mitochondria biogenesis and, in this way, promoting thermogenesis in murine BAT and WAT [79,80,87]. There is also data suggesting that AMPK, by decreasing the DNA methylation of the PRDM16 regulatory region, promotes brown adipocyte differentiation in vitro and stimulates brown adipogenesis and BAT development in animal models [135]. Moreover, chronic activation of AMPK by 5-aminoimidazole-4-carboxamide-1-β-D-ribofuranoside (AICAR) is associated with energy dissipation in cell cultures white adipocytes; however, the exact molecular mechanism of this effect remains unknown [136]. In animal models, AMPK activation with cordycepin, a natural derivative of adenosine, increased energy expenditure, inhibited weight gain, increased WAT browning, and decreased WAT mass enhancing cold tolerance in normal and high-fat diet-fed mice [137]. In turn, selective AMPK knockout in brown and beige adipocytes results in an impairment in mitochondrial structure and function. This leads to resistance to β-adrenergic activation of BAT and subsequently to cold intolerance, insulin resistance, glucose intolerance, and liver steatosis in experimental animals [138].

Interestingly, stimulation of AMPK with subsequent increase of nitric oxide concentrations occurred to be a mechanism by which liraglutide (glucagon-like peptide 1 receptor agonist), a drug used for diabetes and obesity treatment in humans, induces WAT browning and increases energy expenditure and mitochondrial biogenesis in BAT in animal models of obesity [139]

### 5.4. Sirtuins Activators

Sirtuins have also been proposed to stimulate BAT activity and WAT browning. Upon non-specific stimulation (e.g., by cold exposure) and by specific activators (e.g., direct such as RSV and indirect such as capsaicin), SIRT1 deacetylates PPARγ that in the mechanisms described above, promotes transcription of BAT specific genes with parallel repression of genes specific of WAT. Additionally, SIRT1 via activation of PGC-1α promotes transcription of SIRT3, which is crucial for mitochondrial biogenesis and metabolism [140]. What is more, SIRT6, via interaction with ATF2 activates PGC-1α expression. Subsequently, SIRT6 overexpression stimulates the thermogenic program in primary adipocytes, while its silencing reduces the expression of thermogenic genes in brown adipocytes, causing a morphological “whitening” of brown fat, reduced cellular respiration, and obesity [141]. Subsequently, synthetic sirtuins activators have been tested for their efficacy in the stimulation of BAT activation and WAT browning in vivo. Compounds named SRT501 and SRT1720 occurred to increase PGC-1α and enhance oxidative capacity in skeletal muscle, liver, and BAT associated with higher metabolic rate and reduced weight gain [142]. In turn, activation of SIRT6 expression was crucial for thermogenic effectiveness and induction of WAT browning of selective ADRB3 activator CL316 243 since SITRT6 deficiency inhibited WAT browning induced by [141].

### 5.5. SGLT2 Inhibitors

Sodium-glucose cotransporter 2 inhibitors (SGLT2i) are antidiabetic drugs that exert a hypoglycaemic effect due to increased urinary glucose excretion. Since the decrease in glucose load leads to the negative energy balance, SGLT2i treatment is associated with weight reduction. However, recently a novel anti-obesity mechanism of SGLT2i has been proposed. Treatment with empagliflozin resulted in a reduction of M1 pro-inflammatory macrophage infiltration and an increase in anti-inflammatory M2 macrophage population in AT with subsequent enhancement of WAT browning and attenuation of weight gain and insulin resistance [143].

### 5.6. MicroRNAs

MicroRNAs (miRNAs) constitute a class of non-coding, endogenous, small RNA regulating gene expression by translational repression. miRNAs by binding to the complementary sites in the target mRNA cause translation repression, cleavage, deadenylation, and degradation of target mRNA. In recent years there has been significant progress in understanding the role of miRNA in regulating the expression of various genes, including these related to adipocyte differentiation and function. miRNAs implicated in adipogenesis and adipocyte metabolism were found to be differentially expressed in AT from obese subjects before and after weight-loss and normal-weight controls and different AT depots [144]. miRNAs have also been demonstrated to regulate the expression of critical genes involved in WAT browning and BAT thermogenesis. For instance, exposure to cold was found to increase the expression of miR-455 in AT of experimental animals. Subsequently, adipose-specific miR-455 transgenic mice are characterized by marked browning of WAT upon cold exposure. The molecular mechanism involved in this phenomenon includes miR-455-dependent activation of the hypoxia-inducible factor 1 *α* subunit inhibitor (HIF1an) responsible for the stimulation of AMPK [145]. Moreover, the serum concentration of some microRNA (e.g., exosomal miR-92a) inversely correlates with human BAT activity as measured by ^18^F-FDG-PET/CT and could be used as a potential serum biomarker for BAT function in mice and humans in the future [146]. Since a growing number of reports suggest a significant utility of miRNAs as biomarkers for pathogenic conditions and as drugs for medical intervention, one can suspect that targeted activation of WAT browning can constitute a therapeutic option for obesity treatment in the future.

### 5.7. Other Potential Therapeutic Strategies to Enhance WAT Browning

Several other molecular targets have been proposed as potential regulators of BAT activity and/or WAT browning. These include, for instance, cell death-inducing DNA fragmentation factor-α-like effector A (CIDEA). This protein, by suppressing UCP1, increases mitochondrial coupling, and its knockout leads to higher metabolic rate, increased lipolysis in BAT, and resistance to diet-induced obesity and diabetes [147]. Interestingly, CIDEA expression in AT was inversely associated with metabolic rate in humans [148]. Another example is the abovementioned FGF21, which, due to its involvement in WAT browning and oxidation of fatty acids, seems to be a promising therapeutic target for metabolic disease. However, the exact molecular mechanisms of FGF21 action, including its enhancers, counteractors, and downstream signaling pathways, are still unknown. All these are essential to developing FGF21-targeted therapeutic strategies [149]. Bone morphogenetic proteins (BMPs) are other factors that play a regulatory role in adipogenesis and may constitute a potential “grip point” for the pharmacological induction of WAT browning. BMP4 was found to induce differentiation of murine APCs toward the white adipocytes, while BMP7—to promote brown adipogenesis and increase mitochondrial density in brown adipocytes. However, in primary human APCs isolated from subcutaneous WAT, both these proteins occurred to activate PPARγ and subsequently—induce expression of thermogenic genes with a parallel decrease in the white-specific marker levels. However, exposure to BMP4 and BMP7 did not influence mitochondrial content and oxygen consumption in human ASCs [150].

In cultured adipocytes obtained from mice with NPR-C knockout, treatment with ANP led to lipolytic response and increased expression of brown adipocyte markers, while BNP infusion induced WAT browning in vivo. Interestingly, an increase in NP levels constitutes a physiological response to cold exposure, leading to enhanced expression of *npra* in mouse WAT and BAT that favors lipolysis and thermogenesis [41]. In summary, pre-clinical studies suggest that NPs alone or acting synergistically with catecholamines may serve as a therapeutic tool for obesity treatment. However, the challenge may be to determine the appropriate doses and methods of administration so that NPs do not lead to cachexia, frequently seen in patients with advanced heart failure [44]. In turn, treatment with orexin was found to reverse age-related morphologic abnormalities in BAT (reflected e.g., by the increase in the number of multilocular brown adipocytes that have higher lipolytic activity) and to increase UCP1 expression that resulted in improved cold tolerance, glucose homeostasis and insulin sensitivity in experimental animals [46].

Apart from noradrenaline and thyroid hormones, parathyroid hormone (PTH) and PTH-related peptide (PTHRP) demonstrate significant browning potential. This finding comes from the animal models of chronic kidney failure, which is a common cause of secondary hyperparathyroidism. High PTH levels occurred to promote cachexia through induction of WAT browning what can be prevented by a selective PTH receptor knockout in AT [151]. Increased UCP1 expression and enhanced cellular respiration were also found after PTH stimulation in human primary adipocytes isolated from subcutaneous fat [152]. Development of PTH analogs that activate WAT browning without inducing systemic hypercalcemia could, therefore, be a therapeutic strategy for the treatment of obesity.

The described above pharmacological approaches aiming at WAT browning and BAT activation are summarized in Table 2.

## 6. Conclusions

The obesity pandemic has now grown to such proportions that there is a constant need to develop new therapeutic strategies to restore the energy balance. Research of recent years has significantly increased our understanding of the mechanisms related to the development and activation of brown adipose tissue. On the one hand, they explained how factors are known for years, such as cold exposure, physical activity, or specific foods that enhance browning of WAT and activation BAT. They also allowed for the identification of critical molecules and critical steps of both processes setting, therefore, many new therapeutic targets. Most of the proposed WAT browning strategies are confirmed in in vitro models on cell lines as well as in animal models of genetically determined and/or diet-induced obesity. Unfortunately, to date, few of these strategies have found application in inducing WAT browning and BAT activation in humans. Among the probable causes of these discrepancies, one should first consider differences in the mechanisms regulating thermogenesis and its role in the body’s energy expenditure in humans and rodents. Animal models also suggest that WAT depots can differ in their ability to browning; therefore, identifying the distinct cell populations in human WAT depots that can give rise to thermogenically active beige fat cells can be crucial to the development of successful strategies. Moreover, some cellular pathways involved in WAT browning and BAT activation are not adipose tissue exclusive. Therefore, before they become therapeutic targets, their global blockade/activation and potential effects should be considered. Finally, it might occur that in humans, the increased thermogenesis is insufficient to reduce body weight since compensatory mechanisms aimed at energy preservation and increased energy intake can be triggered and a new, higher energy-flux established, probably at higher body mass.

## Figures and Tables

**Figure 1 ijms-21-06241-f001:**
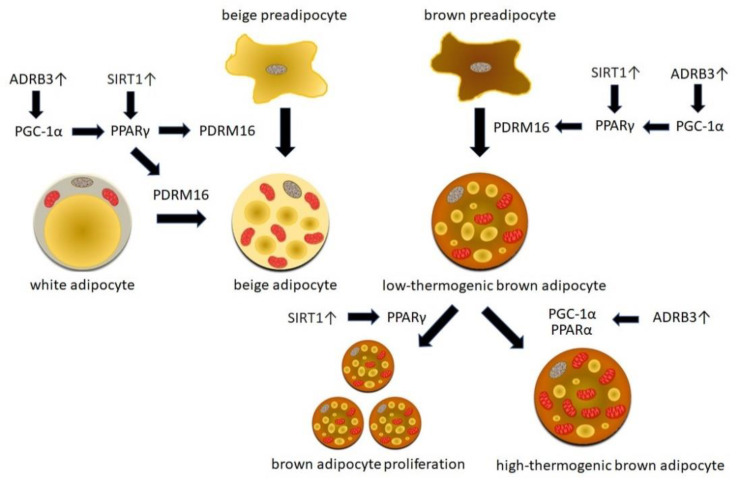
Selected molecular mechanisms involved in white adipose tissue browning and brown adipose tissue activation that could constitute potential therapeutic targets for obesity treatment. Stimulation of adrenergic receptors β3 (ADRB3) is crucial for the initiation of thermogenic pathways, leading to the induction of proliferator-activated receptor gamma (PPARγ) coactivator 1α (PGC-1α) and PPARγ. Sirtuin 1 (SIRT1) also activates PPARγ and enables the recruitment of PRDM16 (PR domain containing 16) transcription factor and initiation of the brown fat specific program. These mechanisms regulate the differentiation of beige progenitors toward mature beige adipocytes and the transdifferentiation of white to beige adipocytes. Stimuli activating ADRB3 and SIRT1 are also involved in brown adipocyte differentiation, proliferation, and activation leading to the change from the low-thermogenic toward the high-thermogenic phenotype.

**Table 1 ijms-21-06241-t001:** Selected non-pharmacological interventions aiming at white adipose tissue (WAT) browning and brown adipose tissue (BAT) activation.

Intervention	Mechanism of Action	WAT	BAT	Experimental Model	Reference
**Cold exposure**					
	↑ ADRB3		↑ brown adipocytes ↑ activity	129Sv mice	[56]
	↑ ADRB3	↑ beige adipocytes		clinical study	[60,61,62]
capsaicin and cold exposure	↑ ADRB3	↑ beige adipocytes		C57BL/6 mice on high-fat diet	[65]
		↓ white adipocytes	↑ activity	clinical study	[67]
**Diet**					
caloric restriction	↑ SIRT1	↑ beige adipocytes		mice on caloric restriction	[111]
capsaicin	↑ ADRB3 ↑ TRPV ↑ SIRT1	↑ beige adipocytes	↑ brown adipocytes ↑ activity	normal-weight rats	[64]
			↑ activity	clinical study	[66]
resveratrol	↑ SIRT1		↑ activity	mice on high-fat diet	[69]
	↑ AMPK	↑ beige adipocytes		WAT-derived murine stromal vascular cells	[70]
				clinical study	[74]
	↑ mTOR		↑ brown adipocytes	3T3-L1 adipocytes	[71]
curcumin	↑ AMPK	↑ beige adipocytes		murine primary white adipocytes 3T3-L1 adipocytes	[78] [78]
				normal-weight mice	[79]
				C57BL/6 mice on high-fat diet	[80]
			↑ activity	C57BL/6 mice on high-fat diet	[80]
green tea catechins	↑ PPARγ	↓white adipocytes	↑ activity	rats on normal diet	[81]
			↑ brown adipocytes	clinical study	[83]
berberine	↑ AMPK		↑ brown adipocytes	obese db/db mice	[87]
			↑ brown adipocytes	clinical study	[88]
PUFA	↑ ADRB3	↑ beige adipocytes		murine white adipocytes	[89]
	↑ TRPV			human white adipocytes	[90]
	↑ FFAR4		↑ activity	rats on high-fat diet	[68]
ATRA	↑ RAR	↑ beige adipocytes	↑ brown adipocytes	murine white adipocytes	[94]
		=	=	human white adipocytes	[94]
		↑ beige adipocytes	↑ brown adipocytes	obese rodents	[95]
**Physical activity**					
	↑ irisin	↑ beige adipocytes		murine white adipocyte	[97]
			↑ brown adipocytes	C57BL/6 mice on high-fat diet	[97]
			↑ activity	C57BL/6 mice on high-fat diet	[97]
	=irisin	=beige adipocytes		clinical study	[96]
	↑ IL-6	↑ beige adipocytes		murine white adipocyte	[99]
		↑ beige adipocytes		mice on high-fat diet	[99]
	↑ FGF21	↑ FFA oxidation	↑ activity	mice with diet-induced obesity	[100]
			↑ activity	clinical study	[102]
**Microbiome**					
	↑ type 2 cytokine signaling	↑ beige adipocytes		obese *ob*/*ob* mice	[106]
				mice on high-fat diet	[106]
	↑ lactate	↑ beige adipocytes		murine and human white adipocyte	[113]
**Bariatric surgery**					
	↑ bile acids	↑ beige adipocytes	↑ activity	clinical studies	[117]
	microbiota	↑ beige adipocytes	↑ activity	clinical studies	[117]

↑ increase/enhancement; ↓ decrease/reduction; ADRB3—adrenergic receptor beta 3; AMPK—AMP-activated protein kinase; ATRA—all-trans retinoic acid; FFA—free fatty acids, FFAR4—free fatty acids receptor 4; FGF21—fibroblast growth factor 21; IL-6—interleukin 6, mTOR—mammalian target of rapamycin; PPARγ—proliferator-activated receptor gamma; PUFA—polyunsaturated fatty acids; SIRT1—sirtuin 1; TRPV—transient receptor potential vanilloid.

**Table 2 ijms-21-06241-t002:** Selected pharmacological approaches aiming at white adipose tissue (WAT) browning and brown adipose tissue (BAT) activation.

Intervention	Compound	Mechanism of Action	WAT	BAT	Experimental Model	Ref
**ADRB3 stimulation**	CL-316243	↑ PGC-1α	↑ lipolysis	↑ brown adipocytes ↑ activity	C57BL/6 mice	[19]
	isoprenaline		=	=	clinical study	[121]
	mirabegron		=	↑ activity	clinical study	[123]
**PPARγ activation**	rosiglitazone	↑ PRDM16	↑ beige adipocytes	↑ activity	C57BL/6 mice	[2]
			↑ beige adipocytes		C57BL/6 mice	[125]
	GQ-16		↑ beige adipocytes	↑ brown adipocytes ↑ activity	mice on high-fat diet	[126]
	imatinib		↑ beige adipocytes		C57BL/6 mice	[128]
	pioglitazone			↓ brown adipocytes ↓ activity	clinical study	[130]
**PPARα activation**	fenofibrate	↑ PRDM16	↑ beige adipocytes		mice with diet-induced obesity	[131]
	pemafibrate		↑ beige adipocytes	↑ activity	mice with diet-induced obesity	[132]
**AMPK activation**	A-769662	↑ PGC-1α	↑ beige adipocytes	↑ activity	obese *ob*/*ob* mice	[134]
			↑ beige adipocytes	↑ activity	AMPK KO mice	[35]
	AICAR	?	↑ energy dissipation		white rat adipocytes	[136]
	cordycepin	?	↑ beige adipocytes	↑ activity	mice on high-fat diet	[137]
	liraglutide	↑ NO	↑ beige adipocytes	↑ activity	mice on high-fat diet	[139]
**SIRT1 activation**	resveratrol	↑ PPARγ	↑ beige adipocytes	↑ activity	mice on high-fat diet and 3T3-L1 adipocytes	[69]
		↑ AMPK	↑ beige adipocytes		WAT-derived murine stromal vascular cells	[70]
	SRT501, SRT1720	↑ PGC-1α		↑ activity	mice on high-fat diet	[142]
**SGLT2i**	empagliflozin	↑ M2 macrophages ↓ M2 macrophages	↑ beige adipocytes		mice with diet-induced obesity	[143]
**MicroRNAs**	miR-455	↑ HIF1an	↑ beige adipocytes	↑ brown adipocytes	human white adipocytes	[145]
	miR-92a	?		↓ activity	clinical study	[146]

↑ increase/enhancement; ↓ decrease/reduction; ? unknown mechanism; ADRB3—adrenergic receptor beta 3; AICAR—5-aminoimidazole-4-carboxamide-1-β-D-ribofuranoside; AMPK—AMP-activated protein kinase; HIF1an—hypoxia inducible factor 1 *α* subunit inhibitor; NO—nitric oxide; PPAR—proliferator-activated receptor; PGC-1α—PPARγ coactivator 1α; PRDM16—PR domain containing 16 transcription factor; SGLT2i—sodium-glucose cotransporter 2 inhibitors; SIRT1—sirtuin 1.

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
