# Peer review of "Induction of Adipose Tissue Browning as a Strategy to Combat Obesity"

_ijms, 2020, doi:10.3390/ijms21176241_

Round 1

Reviewer 1 Report

The manuscript of Kurylowicz and  Puzianowska-Kuznicka, covers some relevant aspects on modulation of adipose tissue browning as a strategy to combat obesity. The review is well written. However,  major revision is required in order the paper may be suitable of publication. In particular,  aspects on the pathogenesis of obesity and the inflammatory mechanisms involved in the course of the disease should be better addressed. In addition, the role of several molecular targets that could play an important role in obesity, such as JNK and NF-kB, must be expanded with specific paragraphs. Finally, the relevance of some molecules already known  for their possible effects on obesity and browning, such as GABA, BAIBA, as well as other polyphenols (e.g. resveratrol and curcumin) should be better assessed.

Author Response

Reviewer 1

  • The manuscript of Kurylowicz and  Puzianowska-Kuznicka, covers some relevant aspects on modulation of adipose tissue browning as a strategy to combat obesity. The review is well written. However,  major revision is required in order the paper may be suitable of publication.

We thank the Reviewer for the positive reception of our manuscript and drawing attention to some weak points, we tried to improve.

  • In particular, aspects on the pathogenesis of obesity and the inflammatory mechanisms involved in the course of the disease should be better addressed.

We do agree with the Reviewer that a chronic inflammatory state that accompanies obesity (known as metaflammation) plays a central role in its progression and development of related metabolic complications. Since we discussed these issues in detail in our recent work (Molecules. 2020;25(9):2224.) now, we focused on those aspects of metaflammation that are related to the induction of adipose tissue browning and/or the activation of thermogenesis in BAT, adding a new section to the revised version of the manuscript with proper citations.

3.2. Role of metaflammation in the regulation of WAT browning and BAT activity.

“Both in humans and experimental animals, obesity is associated with permanently elevated serum levels of inflammatory markers. This chronic, low-grade inflammatory state associated with excessive adiposity is called metaflammation. The pathogenesis of metaflammation is composed and still an object of investigations. Briefly, overnutrition leads to the excess of free fatty acids (FFA) in the circulation that, together with the accumulation of lipids in tissues, cause profound changes in cell metabolism. These include, among others, mitochondrial dysfunction, endoplasmic reticulum stress, and hypoxia resulting in increased expression of genes encoding cytokines, chemokines, and adhesion molecules that subsequently attracts infiltrating immune cells that additionally contribute to the synthesis of pro-inflammatory mediators impairing tissue function [48]. Even though WAT is believed to be a primary site of metaflammation, the inflammatory process related to obesity (assessed by the intensity of macrophages infiltration and concentration of pro-inflammatory cytokines) also affects BAT, as it was shown in animal models [49,50]. Inflammatory mediators impair BAT function that concern also its thermogenic activity. In genetically obese ob/ob mice and mice with diet-induced obesity, high concentrations of inflammatory markers (like e.g., tumor necrosis factor α or IL-1β) inhibit activation of PPARγ, leading to the decreased expression of UCP1, that translates to impaired cold-induced thermogenesis. Interestingly, this process can be reversed by the depletion of pro-inflammatory macrophages [51]. Moreover, it was shown that activation of toll-like receptors (TLRs, that by recognition of conserved pathogen-associated molecular patterns represent the first line of defense against infections), or direct administration of pro-inflammatory cytokines, can repress induction of WAT browning induced by activation of ADRBs [reviewed in 52].” Page 6, lines 239-260

“Inflammation also impairs sympathetic signaling in BAT. The suggested molecular mechanisms underlying this phenomenon include, among others, reduced cAMP synthesis due to the chronic activation of NF-κB cascade and increased clearance of norepinephrine by the obesity-specific population of neuron-associated macrophages. Both result in repressed BAT thermogenic activity and WAT browning [56]. In summary, even though WAT is the primary side of metaflammation, the chronic inflammatory state has deleterious effects on the thermogenic function of BAT. Moreover, local pro-inflammatory signaling in white adipose tissue impairs its browning.” Page 6, lines 275-282

Besides, in sections concerning the non-pharmacological and pharmacological methods of induction of WAT browning and BAT activation, we pointed out that some of the methods discussed also have a favorable influence on the inflammatory state, e.g.:

“Moreover, upregulation of SIRT1 by RSV results in repression of NF-κB pro-inflammatory responses in adipocytes and macrophages that infiltrate adipose tissue, leading to an improvement in insulin signaling and insulin resistance, as it was shown in genetically obese db/db mice [74].” Page 9, lines 363-366

“Administration of curcumin, by arresting lipogenesis in the liver and the inflammatory pathways in adipocytes, prevents the development of high-fat diet-induced obesity and insulin resistance in rodents [77]. (…) Moreover, treatment with curcumin reduced macrophages infiltration and proinflammation cytokine expression in inguinal WAT, suggesting its potential to counteract obesity-associated inflammation [80,81].” Page 9 lines 375-377 & 393-395

“Exercise and physical activity have a well-established role in the preservation of cardiometabolic health. Physical activity also elicits many benefits in AT. These include a reduction in adipocyte size and AT inflammation (e.g., by downregulation of the TLR4 ligation and subsequent activation of NF-κB pathway) as well as an increase in vascularity and mitochondrial biogenesis that results in the improvement of oxidative capacity.” Page 10 lines 448-452

“Moreover, cold, exercise, and caloric restriction can significantly reduce the expression of the critical bacterial enzymes necessary for the lipid A biosynthesis, a critical component of lipopolysaccharide (LPS). Decreased LPS production led to anti-inflammatory M2 macrophage polarization and eosinophil infiltration in WAT, associated with the thermogenic reprogramming of WAT [112,115].” Page 12 lines 525-529

“Treatment with empagliflozin resulted in a reduction of M1 pro-inflammatory macrophage infiltration and an increase in anti-inflammatory M2 macrophage population in AT with subsequent enhancement of WAT browning and attenuation of weight gain and insulin resistance [145].” Page 18 lines 674-677

Hotamisligil, G.S. Inflammation, metaflammation, and immunometabolic disorders. Nature 2017, 542, 177–185.

Sakamoto, T.; Nitta, T.; Maruno, K.; Yeh, Y.S.; Kuwata, H.; Tomita, K.; Goto, T.; Takahashi, N.; Kawada, T. Macrophage infiltration into obese adipose tissues suppresses the induction of UCP1 level in mice. Am J Physiol Endocrinol Metab 2016, 310, E676–E687.

Roberts-Toler, C.; O’Neill, B.T.; Cypess, A.M. Diet-induced obesity causes insulin resistance in mouse brown adipose tissue. Obesity (Silver Spring) 2015, 23, 1765–1770.

Nøhr, M.K.; Bobba, N.; Richelsen, B.; Lund, S.; Pedersen, S.B. Inflammation Downregulates UCP1 Expression in Brown Adipocytes Potentially via SIRT1 and DBC1 Interaction. Int J Mol Sci 2017,18, 1006. 

Villarroya, F.; Cereijo, R.; Gavaldà-Navarro, A.; Villarroya, J.; Giralt, M. Inflammation of brown/beige adipose tissues in obesity and metabolic disease. J Intern Med 2018, 284, 492-504.

Pirzgalska, R.M.; Seixas, E.; Seidman, J.S.; Link, V. M.; Sánchez, N. M.; Mahú, I.; Mendes, R.; Gres, V.; Kubasova, N.; Morris, I.; Arús, B. A.; Larabee, C. M.; Vasques, M.; Tortosa, F.; Sousa, A. L.; Anandan, S.; Tranfield, E.; Hahn, M. K.; Iannacone, M.; Spann, N. J.; Glass, C.K.; Domingos, A.I. Sympathetic neuron-associated macrophages contribute to obesity by importing and metabolizing norepinephrine. Nat Med 2017, 23, 1309–1318.

  • In addition, the role of several molecular targets that could play an important role in obesity, such as JNK and NF-kB, must be expanded with specific paragraphs.

Following the Reviewer's suggestion in the revised version of the manuscript, we added fragments regarding the role of JNK and NF-κB in the development of metaflammation.

“The nuclear factor κB (NF-κB) cascade is a chief intracellular signaling pathway involved in the development of inflammatory response in the adipose tissue. The NF-κB family of transcription factors consists of five members: NF-κB1, NF-κB2, RelA, RelB, and c-Rel that act as homo- and heterodimers. NF-κB dimmers are sequestered in the cytoplasm due to their association with κB inhibitors (IκBs). Various stimuli, including those transmitted by TLRs, cause phosphorylation of IκBs and subsequent dissociation from NF-κB. Dissociation of IκB exposes the nuclear localization signal in NF-κB transcription factors and leads to their translocation to the nucleus. There, NF-κB binds to the promoters of target genes, including those involved in the inflammatory response: cytokines, chemokines, adhesion molecules, and many others that inhibit browning of WAT [53].” Page 6, lines 261-269

Bae, J.; Chen, J.; Zhao, L. Chronic activation of pattern recognition receptors suppresses brown adipogenesis of multipotent mesodermal stem cells and brown pre-adipocytes. Biochem. Cell Biol 2015, 93, 251–261.

”Both FFA and the pro-inflammatory cytokines can activate c-Jun N-terminal kinase (JNK), leading to increased phosphorylation of interferon regulatory factor 3 (IRF3) and in this way contributing to the reduced UCP1 expression and WAT browning [54]. Moreover, activated JNK exacerbates metaflammation by increasing expression of pro-inflammatory cytokines in adipocytes and polarization of macrophages toward the M1 pro-inflammatory phenotype [reviewed in 55].” Page 6, lines 270-274

Lucchini, F. C.; Wueest, S.; Challa, T. D.; Item, F.; Modica, S.; Borsigova, M.; Haim, Y.; Wolfrum, C.; Rudich, A.; Konrad, D. ASK1 inhibits browning of white adipose tissue in obesity. Nat Commun. 2020, 11, 1642.

Yun, J.H.M.; Giacca, A. Role of c-Jun N-terminal Kinase (JNK) in Obesity and Type 2 Diabetes. Cells 2020, 9, 706.

  • Finally, the relevance of some molecules already known for their possible effects on obesity and browning, such as GABA, BAIBA, as well as other polyphenols (e.g. resveratrol and curcumin) should be better assessed.

Following the Reviewer's suggestion, in the revised version of the manuscript, we added new sections regarding the role of GABA and BAIBA in the regulation of adipose tissue browning.

“While adrenergic stimulation plays a central role in the induction of adipose tissue browning, and activation of thermogenesis, activation of gamma-aminobutyric acid (GABA) signaling exerts opposite effects in adipocytes. The influence of GABA on energy homeostasis is composed. On the one hand, the activation of the GABA-A receptor in the hypothalamus suppresses food intake and reduces body weight [38]. On the other, the Agouti-related peptide (AgRP) and neuropeptide Y (NPY) neurons – both stimulating appetite, are mostly GABAergic [39]. In the context of brown adipocytes, the constitutive activation of the GABA-BR1 receptor in obese mice leads to excessive calcium influx into the mitochondria causing mitochondrial calcium overload and oxidative stress that is accompanied by low UCP1 expression. Moreover, in experimental animals, high fat diet leads to a chronic up-regulation of the GABA-BR1 in adipose tissue and its translocation into cellular membrane, making adipocytes of obese animals more sensitive to the damaging effects of GABA. In turn, suppression of excessive signaling by the inhibition of mitochondrial calcium overload restores normal BAT function [40].” Page 5, lines 200-212

Turenius, C.I.; Htut, M.M.; Prodon, D.A.; Ebersole, P.L.; Ngo, P.T.; Lara, R.N.; Wilczynski, J.L.; Stanley, B.G. GABA(A) receptors in the lateral hypothalamus as mediators of satiety and body weight regulation. Brain Res 2009, 1262, 16-24.

Tong, Q.; Ye, C.P.; Jones, J.E.; Elmquist, J.K.; Lowell, B.B. Synaptic release of GABA by AgRP neurons is required for normal regulation of energy balance. Nat Neurosci 2008, 11, 998-1000.

Ikegami, R.; Shimizu, I.; Sato, T.; Yoshida, Y.; Hayashi, Y.; Suda, M.; Katsuumi, G.; Li, J.; Wakasugi, T.; Minokoshi, Y.; Okamoto, S.; Hinoi, E.; Nielsen, S.; Jespersen, N.Z.; Scheele, C.; Soga, T.; Minamino, T. Gamma-Aminobutyric Acid Signaling in Brown Adipose Tissue Promotes Systemic Metabolic Derangement in Obesity. Cell Rep 2018 24, 2827-2837.e5.

“Also β-aminoisobutyric acid (BAIBA), via activation of PPAR, occurred to act as an inductor of WAT browning both in vitro and in vivo. BAIBA is considered a myokine since it is a product of valine catabolism generated in the muscle during exercise. In epidemiological studies, plasma valine concentration correlates with obesity and hyperinsulinemia. Therefore its utilization in working muscle can be partially responsible for the beneficial metabolic effects of exercise. Subsequently, in genetically obese mice (ob/+), treatment with BAIBA led to the reduction of weight gain that was accompanied by the improvement of glucose tolerance, enhanced fatty acid oxidation, and reduced lipogenesis in the liver [105].” Page 11, lines 481-488

Roberts, L.D.; Boström, P.; O'Sullivan, J.F.; Schinzel, R.T.; Lewis, G.D.; Dejam, A.; Lee, Y.K.; Palma, M.J.; Calhoun, S.; Georgiadi, A.; Chen, M.H.; Ramachandran, V.S.; Larson, M.G.; Bouchard, C.; Rankinen, T.; Souza, A.L.; Clish, C.B.; Wang, T.J.; Estall, J.L.; Soukas, A.A.; Cowan, C.A.; Spiegelman, B.M.; Gerszten, R.E. β-Aminoisobutyric acid induces browning of white fat and hepatic β-oxidation and is inversely correlated with cardiometabolic risk factors. Cell Metab 2014, 19, 96-108.

Additionally, we expanded fragments concerning the role of polyphenols (including resveratrol and curcumin) in BAT development and metabolism. However, we would like to point out that the reason why we did not describe the role of polyphenols in adipose tissue browning and obesity treatment in more detail, is that we discussed these issues in our recently published review [Nutrients. 2020;12(2):582.]

“Another representative of dietary compounds with thermogenic potential is resveratrol (3,5,4’-trihydroxy-trans-stilbene, RSV), a polyphenol present in grapes, berries, peanuts, and selected teas. The underlying mechanism includes activation of SIRT1, that by direct deacetylation of PPARγ recruits BAT program coactivator PRDM16 resulting in induction of genes typical for BAT and downregulates genes that are specific for WAT in 3T3-L1 preadipocytes [70]. Moreover, the addition of RSV to a cell culture of differentiated inguinal WAT-derived stromal vascular cells significantly increased brown and beige adipocyte markers' expression. This effect of resveratrol on beige adipocyte formation is partially mediated by phosphorylation of AMPK since AMPK inhibition eliminates the browning effects of RSV on WAT [71]. Besides, a recent study pointed on the mammalian target of the rapamycin (mTOR) pathway to be involved in RSV induced WAT browning [72].

In animal models of obesity, the addition of RSV to a high-fat diet-induced UCP1 expression and oxygen consumption in BAT, resulting in the increased basic metabolic rate and, in this way, successfully counteracted accumulation of AT [73]. Moreover, the upregulation of SIRT1 by RSV results also in the repression of NF-κB pro-inflammatory responses in adipocytes and macrophages that infiltrate adipose tissue, leading to an improvement in insulin signaling and insulin resistance, as it was shown in genetically obese db/db mice [74]. Interestingly, a four-week treatment with 500 mg of trans-resveratrol led to the upregulation of thermogenesis-related genes, also in human WAT [75]. This finding can be partially responsible for the beneficial influence of RSV and its derivates on weight management observed in clinical trials [reviewed in 76].

 What is important, after ingestion, RSV is metabolized in the colon by the gut microbiota, and part of the beneficial effects of RSV is mediated via its microbial metabolites. Therefore, depletion of the gut microbiota may counteract the beneficial effects of RSV on brown adipose tissue and glucose metabolism [74].” Pages 8-9, lines 350-373

Baur, J.A.; Pearson, K.J.; Price, N.L.; Jamieson, H.A.; Lerin, C.; Kalra, A.; Prabhu, V.V.; Allard, J.S.; Lopez-Lluch, G.; Lewis, K.; Pistell, P.J.; Poosala, S.; Becker, K.G.; Boss, O.; Gwinn, D.; Wang, M.; Ramaswamy, S.; Fishbein, K.W.; Spencer, R.G.; Lakatta, E.G.; Le Couteur, D.; Shaw, R.J.; Navas, P.; Puigserver, P.; Ingram, D.K.; de Cabo, R.; Sinclair, D.A. Resveratrol improves health and survival of mice on a high-calorie diet. Nature 2006, 444, 337–342.

Hui, S.; Liu, Y.; Huang, L.; Zheng, L.; Zhou, M.; Lang, H.; Wang, X.; Yi, L.; Mi, M. Resveratrol enhances brown adipose tissue activity and white adipose tissue browning in part by regulating bile acid metabolism via gut microbiota remodeling. Int J Obes (Lond) 2020, 44, 1678-1690.

”Curcumin is a naturally occurring curcuminoid of turmeric, with anti-obesity and anti-hyperglycemic properties. Administration of curcumin, by arresting lipogenesis in the liver and the inflammatory pathways in adipocytes, prevents the development of high-fat diet induced obesity and insulin resistance in rodents [77]. In turn, in a randomized, controlled study, in overweight subjects with central adiposity and glucose intolerance, adding curcumin to nutritional intervention and physical activity significantly increased weight and body fat reduction [78].

 Moreover, due to the ability to activate PPARγ and AMPK, curcumin also has thermogenic properties. The browning potential of curcumin has been shown in vitro in 3T3-L1 cells and mouse primary white adipocytes [79]. In this study addition of curcumin to cell cultures induced the beige phenotype and drove the BAT thermogenic program through significant upregulation of brown fat-specific genes. Moreover, curcumin increased the number of mitochondria and expression of critical mitochondrial proteins such as carnitine palmitoyltransferase I (CPT1, responsible for the oxidation of fatty acids in brown adipocytes) and cytochrome C that plays a key role in mitochondrial oxidative phosphorylation. Therefore, the anti-obesity properties of curcumin are based on both: ability to inhibit lipogenesis and induction of thermogenesis and adipose tissue browning [79].

 This observation has been confirmed in vivo when curcumin administered 50-100 mg/kg/day decreased body weight and fat mass without affecting food intake but by increasing energy expenditure and body temperature in mice. These curcumin effects were associated with the emergence of beige adipocytes with an increase of brown-specific markers expression and mitochondrial biogenesis in WAT. Moreover, treatment with curcumin reduced macrophages infiltration and proinflammation cytokine expression in inguinal WAT, suggesting its potential to counteract obesity-associated inflammation [80,81]. However, curcumin's effectiveness in the induction of adipose tissue browning in humans has to be established yet.” Page 9, lines 374-396

Shao, W.; Yu, Z.; Chiang, Y.; Yang, Y.; Chai, T.; Foltz, W.; Lu, H.; Fantus, I.G.; Jin, T. Curcumin prevents high fat diet induced insulin resistance and obesity via attenuating lipogenesis in liver and inflammatory pathway in adipocytes. PLoS One 2012, 7, e28784.

Reviewer 2 Report

Comments to the author Alina Kuryłowicz et.al summarized concepts regarding BAT activation and WAT browning in detail. ANP/BNP and orexin should be added in the manuscript.   1) ANP/BNP signaling   J Clin Invest. 2012 Mar;122(3):1022-36. Cardiac natriuretic peptides act via p38 MAPK to induce the brown fat thermogenic program in mouse and human adipocytes   Sci Rep. 2017 Oct 11;7(1):12978. The thermogenic actions of natriuretic peptide in brown adipocytes: The direct measurement of the intracellular temperature using a fluorescent thermoprobe.   Mol Metab. 2018 Mar;9:192-198. Cardiac natriuretic peptides promote adipose 'browning' through mTOR complex-1   2)orexin   Cell Metab. 2011 Oct 5;14(4):478-90. Orexin is required for brown adipose tissue development, differentiation, and function   Endocrinology. 2014 Feb;155(2):485-501. Orexin restores aging-related brown adipose tissue dysfunction in male mice

Author Response

Reviewer 2

ANP/BNP and orexin should be added in the manuscript.

1) ANP/BNP signaling   J Clin Invest. 2012 Mar;122(3):1022-36. Cardiac natriuretic peptides act via p38 MAPK to induce the brown fat thermogenic program in mouse and human adipocytes   Sci Rep. 2017 Oct 11;7(1):12978. The thermogenic actions of natriuretic peptide in brown adipocytes: The direct measurement of the intracellular temperature using a fluorescent thermoprobe.   Mol Metab. 2018 Mar;9:192-198. Cardiac natriuretic peptides promote adipose 'browning' through mTOR complex-1  

Following the Reviewer’s suggestion in the revised version of the manuscript, we have added a new section on the role of ANP/BNP signaling in adipose tissue browning, including proper citations.

“Over the past decade, many other pathways involved in the development of brown adipose tissue and the regulation of thermogenesis have been discovered. The examples are atrial (ANP) and brain (BNP) natriuretic peptides. Both of them, via activation of natriuretic peptide receptors (NPRs) and downstream mediators like cyclic guanosine monophosphate (cGMP), protein kinase G (PKG) and mammalian target of rapamycin complex 1 (mTORC1), play a central role in the regulation of fluid balance and hemodynamic. However, identification of NP receptors (NPRs) expression in adipose tissue shed new light on ANP/BNP pathways' role in the whole body homeostasis. Subsequently, ANP was found to induce lipolysis in white adipocytes and thermogenesis in BAT, while both NPs, via the cGMP-PKG-mTORC1 pathway, can increase UCP1 expression and mitochondrial content in adipocytes leasing to WAT browning [41-43]. Accordingly, mice with npra knockout suffer from cardiac hypertrophy and have significantly higher adipose tissue content than wild-type animals [41]. On the contrary, patients with chronic heart failure that are exposed to persistently elevated NPs levels are often diagnosed with lipodystrophy and cachexia (known as cardiac cachexia) [44].” Page 5, lines 213-226

Additionally, we discussed the effectiveness of strategies aiming at ANP/BNP signaling in the treatment of obesity in animals and the perspective of their application in humans. 

“In cultured adipocytes obtained from mice with NPR-C knockout, treatment with ANP led to lipolytic response and increased expression of brown adipocyte markers, while BNP infusion induced WAT browning in vivo. Interestingly, an increase in NP levels constitutes a physiological response to cold exposure, leading to enhanced expression of npra in mouse WAT and BAT that favors lipolysis and thermogenesis [41]. In summary, pre-clinical studies suggest that NPs alone or acting synergistically with catecholamines may serve as a therapeutic tool for obesity treatment. However, the challenge may be to determine the appropriate doses and methods of administration so that NPs do not lead to the cachexia frequently seen in patients with advanced heart failure [44].” Page 19, lines 717-724

Bordicchia, M.; Liu, D.; Amri, E.Z.; Ailhaud, G.; Dessì-Fulgheri, P.; Zhang, C.; Takahashi, N.; Sarzani, R.; Collins, S. Cardiac natriuretic peptides act via p38 MAPK to induce the brown fat thermogenic program in mouse and human adipocytes. J Clin Invest 2012, 122, 1022-36.

Kimura, H.; Nagoshi, T.; Yoshii, A.; Kashiwagi, Y.; Tanaka, Y.; Ito, K.; Yoshino, T.; Tanaka, T.D.; Yoshimura, M. The thermogenic actions of natriuretic peptide in brown adipocytes: The direct measurement of the intracellular temperature using a fluorescent thermoprobe. Sci Rep 2017, 7, 12978.

Liu, D.; Ceddia, R.P.; Collins, S. Cardiac natriuretic peptides promote adipose 'browning' through mTOR complex-1. Mol Metab. 2018, 9, 192-198.

Okoshi, M.P.; Capalbo, R.V.; Romeiro, F.G.; Okoshi, K. Cardiac Cachexia: Perspectives for Prevention and Treatment. Arq Bras Cardiol 2017, 108, 74-80.

2) orexin  Cell Metab. 2011 Oct 5;14(4):478-90. Orexin is required for brown adipose tissue development, differentiation, and function   Endocrinology. 2014 Feb;155(2):485-501. Orexin restores aging-related brown adipose tissue dysfunction in male mice

Following the Reviewer’s suggestion, we discussed the role of orexin in the development of brown adipose tissue and its possible use to counteract obesity, including proper citations.

“Another examples are orexins (OXs) A and B – neuropeptides that regulate sleep-wake cycles, physical activity, and appetite, but also may increase adipose tissue browning and thermogenesis [45]. Orexins, via activation of MAPK and BMPR1A signaling, are able to initiate the browning program in mesenchymal progenitor stem cells, embryonic fibroblasts, and brown preadipocytes. Subsequently, mice deficient with orexins have impaired differentiation of brown preadipocytes toward mature brown cells and decreased adaptive thermogenesis. Therefore, in response to the high caloric diet, they experience a rapid increase in adipose tissue volume [45]. Interestingly, a decrease in BAT's orexin receptors expression, leading to loss of its thermogenic capacity, was found to be responsible for the aging-associated increase in adiposity in mice [46]. Moreover, low orexin serum levels are associated with obesity in humans [47].” Page 5, lines 227-236

“In turn, treatment with orexin was found to reverse age-related morphologic abnormalities in BAT (reflected e.g., by the increase in the number of multilocular brown adipocytes that have higher lipolytic activity) and to increase UCP1 expression that resulted in improved cold tolerance, glucose homeostasis and insulin sensitivity in experimental animals [46].” Page 19, lines 725-728

Sellayah, D.; Bharaj, P.; Sikder, D. Orexin is required for brown adipose tissue development, differentiation, and function Cell Metab. 2011, 14, 478-490.

Sellayah, D.; Sikder, D. Orexin restores aging-related brown adipose tissue dysfunction in male mice. Endocrinology 2014, 155, 485-501.

Knutson, K.L.; Van Cauter, E. Associations between sleep loss and increased risk of obesity and diabetes. Ann N Y Acad Sci 2008,1129, 287-304.

Round 2

Reviewer 1 Report

The manuscript appears now significaantly improved and merits ans suitable of publication